# Impact of Portal Vein Resection (PVR) in Patients Who Underwent Curative Intended Pancreatic Head Resection

**DOI:** 10.3390/biomedicines11113025

**Published:** 2023-11-11

**Authors:** Markus Bernhardt, Felix Rühlmann, Azadeh Azizian, Max Alexander Kölling, Tim Beißbarth, Marian Grade, Alexander Otto König, Michael Ghadimi, Jochen Gaedcke

**Affiliations:** 1Department of General, Visceral and Pediatric Surgery, University Medical Center, D-37075 Goettingen, Germanymaxalexander.koelling@med.uni-goettingen.de (M.A.K.);; 2Institute of Medical Bioinformatics, University Medical Center, D-37075 Goettingen, Germany; 3Department of Gastroenterology and Gastrointestinal Oncology, University Medical Center, D-37075 Goettingen, Germany

**Keywords:** pancreatic surgery, portal vein resection, pancreatic cancer

## Abstract

The oncological impact of portal vein resection (PVR) in pancreatic cancer surgery remains contradictory. Different variables might have an impact on the outcome. The aim of the present study is the retrospective assessment of the frequency of PVR, histological confirmation of tumor infiltration, and comparison of oncological outcomes in PVR patients. We retrieved *n* = 90 patients from a prospectively collected data bank who underwent pancreas surgery between 2012 and 2019 at the University Medical Centre Göttingen (Germany) and showed a histologically confirmed pancreatic ductal adenocarcinoma (PDAC). While 50 patients (55.6%) underwent pancreatic resection combined with PVR, 40 patients (44.4%) received standard pancreatic surgery. Patients with distal pancreatectomy or a tumor other than PDAC were excluded. PVR was performed either as local excision or circular resection of the portal vein. Clinical/patient data and follow-ups were retrieved. The median follow-up period was 20.5 months. Regarding the oncological outcome, a statistically poorer CSS (*p* = 0.04) was observed in PVR patients. There was no difference (*p* = 0.18) in patients’ outcomes between tangential and complete PVR, while *n* = 21 (42% of PVR patients) showed portal vein infiltration. The correlation between performed PVR and resection status was statistically significant: 48.6% of PVR patients achieved R0 resections compared to 75% in non-PVR patients (*p* = 0.03). Patients who underwent PDAC surgery with PVR show a significantly poorer outcome regardless of PVR type. Tumor size and R-status remain two important variables significantly associated with outcome. Since there is a lack of standardization for the indication of PVR, it remains unknown if the need for resection of vein structures during pancreatic resection represents the biological aggressiveness of the tumor or is biased by the experience of the surgeon.

## 1. Introduction

Until today, surgical resection of pancreatic cancer (which is mostly a pancreatic ductal adenocarcinoma, PDAC) remains the only curative treatment for this disease. Although systemic therapy options have improved over the last years, and especially the benefit of neoadjuvant treatment is discussed frequently, radical oncological resection of pancreatic cancer with the aim of a complete resection (R0) and sufficient lymph node dissection are the center of any curative treatment. Both lymph node status and resection margin status (R0/R1) are known to be strong prognostic factors for survival.

However, not only have the chemotherapy options improved, but developments in surgical techniques and perioperative care have also occurred over the last few years. Resection of portal vein (PV) or superior mesenteric vein (SMV) became the standard of care in pancreatic surgery if necessary to achieve complete resection (R0). In contrast to arterial resections, vein resections can be safely performed without an increase in morbidity or mortality [1,2]. Over the past years, the oncological role of PV/SMV resection was discussed. While some authors consider vein infiltration as a sign of tumor aggressiveness, others reported comparable results in pancreatic surgery between patients with and without PV/SMV resection. Until today, the prognostic role of PV resection (PVR) is still under debate [1,3,4,5,6,7,8,9,10,11] and needs further robust data about whether (and in which cases) patients benefit from PVR to help surgeons make this decision during surgery. However, it should be mentioned that PVR and a histologically confirmed PV infiltration of the tumor are two aspects that need to be addressed separately as well as combined. Whether the PV is infiltrated or not is information surgeons often receive after surgery is performed. During surgery, infiltration of the PV on a cellular level is not detectable.

In a previous analysis [12], we evaluated patients with pancreatic head cancer and a rigid histopathological work-up. With respect to histopathological confirmed infiltration of the PV, no statistically significant survival difference was found. However, we identified a relevant number of patients (*n* = 23) undergoing PVR without histological infiltration of the portal vein. This subgroup revealed a significantly better prognosis. At first, this number appeared surprisingly high, but the range of infiltration may vary between 30 and 80% (or even more), according to a recent meta-analysis by Song et al. [3]. Therefore, retrospective comparisons between infiltrated and non-infiltrated PV might not be quite accurate, considering that in the clinical setting, a rigid histopathological work-up with a clear statement of whether the portal vein is infiltrated or not is often missing. However, it can also be explained by the fact that the decision for (or against) PVR is made individually by the surgeon while performing the surgery. In the case of PV adherence to the pancreatic head, PVR is usually performed. Whether this is made by tangential or circular resection depends on the type of adherence and infiltration, respectively. Also, the type of resection and reconstruction (primary suture or patch in local excision), as well as the end-to-end anastomosis or graft interposition in circular excision [13], plays a separate role, as published by Ravikumar [14]. As another interfering factor, the surgeon performing the resection himself has an important impact on the extension of the procedure and patients’ outcomes. An experienced surgeon might perform a more radical resection. Also, patients’ age and comorbidities might lead to a more or less radical resection.

Besides the mentioned factors, which might alter the outcome of patients undergoing pancreatic surgery with or without PVR, resection of the portal vein itself might have some possible outcome-improving or outcome-compromising effects. One possible theory is that even if the PV itself is not infiltrated, PVR might be necessary to enable complete resection of the infiltrated tissue close to the PV. On the other hand, PVR might cause sinistral (left-sided) portal vein hypertension [15,16], which can result in life-threatening bleeding in the upper gastrointestinal tract.

In the present study, we aimed to further elucidate the role of PVR in terms of frequency, infiltration and prognostic role.

## 2. Materials and Methods

Data from patients with pancreatic head surgery or total pancreatectomy due to pancreatic ductal adenocarcinoma (PDAC) between 2012 and 2019 at the Department of General and Visceral Surgery at the University Medical Center were analyzed. Since PVR is usually not part of a distal pancreatectomy, those patients were excluded from further analysis. For a homogenous cohort, all patients with distal bile duct carcinoma, papillary carcinoma, neuroendocrine tumors or duodenum carcinoma were excluded. To achieve an independent cohort, all patients who were already included in the former study to assess the relevance of an intensified histopathological work-up were excluded.

In total, data from *n* = 90 patients were enrolled in the presented study. Patients and clinical data, as well as follow-up results, were retrieved from a prospectively collected databank. For patients who underwent pancreatic surgery, data on PV resection, the type of resection, as well as tumor infiltration, were retrospectively collected. Follow-up examinations were either performed in-house or data were retrieved from general practitioners or oncologists. CT scans during follow-up were either part of the surveillance or they became necessary if recurrence was suspected. The surgical procedure was performed as previously described [12]. Portal vein resection was performed by either local excision and primary suture or circular resection of PV and end-to-end anastomosis. Venous patches or grafts were not applied.

To assess the relevance of PVR compared to non-PVR in pancreatic head resection, overall survival (OS), cancer-specific survival (CSS) and disease-free survival (DFS) were evaluated, and Kaplan–Meier curves were used for illustration. Gehan–Breslow–Wilcoxon tests were performed for type of PV resection, PV infiltration and oncological relevance of tumor size, accordingly. To finally assess whether patients after PVR show different R0 resection rates, Fisher’s exact test was applied. All graphs and statistical analyses were performed with GraphPad Prism version 8.4.3 for Mac (GraphPad Software, La Jolla, CA, USA). For all statistical analyses, statistical significance was accepted for *p* < 0.05.

The local ethics comity of University Medical Center Goettingen approved the presented study and waived the need for informed consent due to its retrospective character.

## 3. Results

### 3.1. Clinical and Demographic Data

Overall, *n* = 90 patients were included in the analysis. The median age was 71.6 years (range: 31–90 years), *n* = 60 (66.6%) were female, *n* = 30 (33.3%) were male. The vast majority of surgical procedures performed was a pylorus-preserving pancreaticoduodenectomy (PPPD, *n* = 76, 84.4%), followed by total pancreatectomies (*n* = 8, 8.9%) and Whipple resections (*n* = 6, 6.7%). In *n* = 50 patients (55.6%), PV/SMV resection was performed. In *n* = 13 patients (26%), a wedge resection with primary suture was possible, whereas *n* = 37 (74%) needed complete resection with end-to-end anastomosis. None of the included patients showed distant metastases (cM0). All relevant clinical data are shown in Table 1.

### 3.2. Histopathological Work-Up and Patients’ Outcome

To assess the oncological representativeness of our cohort, patients were analyzed according to resection status and tumor size based on the well-established diameter of 3 cm indicated by the pathologist. The median follow-up period was 20.5 months. Here, 77.8% of the patients showed tumor recurrence during the first two years after surgery. The median survival of all patients was 20.4 months. Moreover, 10% of all included patients remained recurrence-free until the end of follow-up.

Patients with small tumor size showed a significantly better prognosis in OS (tumor size < 3 cm vs. tumor size > 3 cm: 29.7 vs. 16.1 months, *p* = 0.03) and in CSS (tumor size < 3 cm vs. tumor size > 3 cm: 34.5 vs. 16.4 months, *p* = 0.008); histopathological complete tumor removal trended towards better outcome in OS (R0 vs. R1: 22 vs. 14.4 months; *p* = 0.063; see Figure 1) but CSS and DFS did not show any statistically significant outcome improvements (CSS: R0 vs. R1).

### 3.3. PVR and Patients’ Outcome

To evaluate patients’ long-term outcomes, overall survival, cancer-specific survival, and disease-free survival were analyzed, comparing PVR and non-PVR patients as well as tangential and complete resection types.

Patients with PVR were found to be significantly associated with poorer cancer-specific survival (CSS; *p* = 0.04, HR = 1.79). Overall survival (OS; *p* = 0.07, HR = 1.61) trended towards a poorer outcome for PVR vs. non-PVR patients without being statistically significant. No prognostic difference was found in disease-free survival (DFS; *p* = 0.1, HR = 1.60; Figure 2A–C). Between the resection types (tangential vs. complete), no statistical difference was seen in the outcome.

Since the resection status is a known prognostic factor for survival, we further analyzed the long-term impact of PVR in patients with R0-state and R1-state separately.

Consequently, four groups were compared: Patients with R0-state without PVR, patients with R0-state with PVR, patients with R1-state without PVR, and patients with R1-state and PVR.

As shown in Figure 3, patients with PVR show a similar CSS regardless of their R-state. Patients with R0-state and without PVR show significantly better CSS compared to all of the other groups.

As mentioned before, another possible prognostic factor is lymph node metastasis (nodal-state). Here, we performed an analysis to compare the long-term impact of PVR dependent on the nodal state. Consequently, four groups were compared: Patients with N0-state without PVR, patients with N0-state with PVR, patients with N1-state without PVR, and patients with N1-state and PVR. No significant differences in survival could be shown between all four groups.

### 3.4. Impact of Tumor Infiltration of the Portal Vein

A further analysis was performed with regard to tumor infiltration of the PV. A certain number of resections do not show any tumor infiltration after histopathological work-up. Reliable data were available for 47 of 50 patients (94%). Twenty-one (44.7%) of them showed a positive infiltration, whereas the majority (*n* = 26, 55.3%) was resected without histopathological confirmed infiltration. With respect to the prognosis, no difference was found between patients with tumor infiltration of the portal vein compared to patients with PVR but without any tumor infiltration. A significant difference was found in the comparison of PV/SMV infiltrated patients and those without any PVR. Patients with infiltration of the portal vein showed a significantly worse OS compared to non-PVR patients (Figure 4; *p* = 0.03; HR = 1.9).

R0 was achieved in *n* = 30 patients of non-PV/SMV resection (75%). Comparable results were found for tangential resection with R0 in *n* = 10 patients (76,9%). Patients undergoing complete vein resection showed a significantly lower rate of R0 (Figure 5; Odds-Ratio = 3.17, *p* = 0.03).

## 4. Discussion

In the present study, we retrospectively analyzed *n* = 50 patients with histopathologically confirmed PDAC who underwent curative pancreatic surgery, including PVR, to *n* = 40 patients with PDAC and pancreatic surgery without PVR. While female patients were slightly overrepresented (66,6%), median age, TNM stage, grading and resections status are in line with the general distribution in PDAC patients. To evaluate the validity of our follow-up data, we compared patients’ outcomes to tumor size and resection status (R). Both results, small tumor size and complete resection (R0) status being associated with better outcomes (see Figure 1), are in line with the existing literature. Overall, the present patient cohort and follow-up data seem to be oncologically representative.

Complete resection is the main goal of pancreatic cancer surgery, as margin positivity (R1) is a prognostic marker for poorer prognosis. In the case of a PV/VMS infiltration, the en-bloc vein resection is a mandatory step. However, the possibility of resection is limited in those cases where tumor infiltration is within the mesenteric root and multiple small branches of the VMS. Until today, it is under debate if vein infiltration and, consequently, its resection is just a necessary step, e.g., due to the localization and size of the tumor or if it is an expression of biological aggressiveness [17,18,19]. In the present study, patients with PVR show significantly poorer prognosis in terms of CSS. However, the rate of R1 resections is also higher in the group of PVR patients, which might add to the effect resulting in poorer prognosis. Also, both aspects taken together (necessity of PVR and R1-status) could support the theory that tumors with adherence to the PV are those with a more aggressive tumor biology spreading locally on a cellular basis. Interestingly, patients with PVR show significantly poorer survival regardless of the R-status (see Figure 3), which might lead to the conclusion that whenever PVR is necessary, the patient will have a poorer prognosis, even if the resection margin is free of tumor cells.

In addition to the resection itself, the actual tumor infiltration of the PV is a main aspect that needs further consideration in this matter. Here, we showed that patients with actual tumor infiltration of the portal vein have a significantly poorer OS compared to those without any PVR (who represent that group of patients without any adherence of the tumor to the PV and, therefore, without any necessity of PVR). However, as mentioned earlier, whether a PVR is necessary is decided by the surgeon. This decision remains subjective and is surely biased by the surgeon’s experience and the patient’s general state of health.

The discrepancy of the existing literature might have several reasons: First, the fraction of PV/VMS resected patients among all included patients differs tremendously in published data. The ESPAC-4 trial [20], for example, reported a vein resection rate of 14%. The French recommendations of vascular resection in pancreatic cancer report 25% in pancreatic head resection up to 50% in total TP [21]. The meta-analyses from Song et al. [3] revealed a frequency of up to 42%, including predominantly pancreaticoduodenectomies, and Oba et al. reported a PVR rate of up to 62.2% [22]. Second and of importance, resected veins are not always infiltrated. This, however, is not recognized in a relevant number of publications, although more than 20 years ago, Nakao et al. [23] already analyzed a cohort of *n* = 101 patients with pancreatic head resection of whom 88% (*n* = 89) received a PV/SMV resection. Here, vein infiltration was classified according to preoperative (or intraoperative) imaging, depth of tumor invasion was recorded, and finally, the occlusion of the portal vein and the existence of a cavernous transformation was associated with a poor prognosis comparable to non-resected patients. This group of non-PV-infiltrating tumor patients also revealed the best prognosis. A prognostic difference was also found in the previous and the presented data set as well as reported by others [5,7]. Though the decision of portal vein resection is often made during surgery due to the adherence of the portal vein to the pancreas, the percentage of infiltration may vary due to the experience and the skills of the surgeon. This also explains the varying amount of histopathological confirmation of tumor infiltration, which is around 40% to 70% [24]. Terasawa et al. showed in a cohort of *n* = 339 patients that different approaches and techniques and, especially preoperative planning and decision for PVR, contribute to the outcome enormously [25]. This needs to be acknowledged, as already pointed out, and may contribute to the conflicting results.

Another relevant issue contributing to the varying data is the type of venous resection. According to the size of the infiltration, a local or circular excision can be performed. Reconstruction is typically performed by primary suture, patch plastic, end-to-end anastomosis or an interposing graft as classified by the ISGPS [13]. This obviously plays an additional role in assessing the relevance of portal vein resection. As shown in this data set, wedge resection goes along with a more favorable outcome compared to segment resections; even if the difference is not statistically significant, a trend can be clearly seen (Figure 2A–C). These data are supported by different previous analyses [8,14,26,27]. Furthermore, the length of the resected vein may contribute to prognosis [28,29]. This aspect has not been analyzed in the present study.

In the present study, the number of incomplete (R1) resections is significantly higher in PVR patients (Figure 4), which is a result supported by data from Kleive et al., who analyzed *n* = 98 patients with pancreaticoduodenectomy [30]. R1 resection and margin involvement in the SMV group were significantly higher in patients who underwent additional venous resection.

Based on the idea that extended tissue resection may be the driver for better prognosis in PV/VMS resection without infiltration, Klein et al. [31] published data on 40 patients with vein resection but without cancer infiltration. These patients were matched to 120 patients without vein resection, which resulted in a significantly poorer survival for the PV/SMV resection group. Consequently, the study results rebutted the initial hypothesis. Our data confirm those findings. As shown in Figure 5, complete resections of PV are even significantly associated with higher R1 rates.

Overall, PVR seems to have a negative impact on the CSS of PDAC patients undergoing surgery and could qualify as an independent prognostic factor among other known factors (e.g., tumor size, R-state, N-state) if confirmed in other studies. However, some aspects are always to be considered concerning PVR: the surgeon’s experience, the patient’s general condition, the type of PVR and tumor infiltration of the PV. These points should underline the difficulty in standardizing portal or superior mesenteric vein resection and may be a possible explanation for discrepant results in the literature and the absence of reproducibility. Although resection of the portal vein is an accepted technique, it should finally be acknowledged that long-term complications like sinistral portal hypertension [15,16,17,32,33,34], which might lead to gastrointestinal bleeding, are controversially discussed [35] and may depend on the surgical aggressiveness. This at least should be kept in mind to avoid unnecessary vein resections even if it is technically doable.

One further aspect that will most likely gain more importance is the application of neoadjuvant treatment and its impact on pancreatic surgery in general. It might change the effect of PVR concerning outcome, but it also might lead to fewer PVR. In the present study, only *n* = 7 patients had neoadjuvant treatment; therefore, a proper analysis of this subgroup is not possible within the presented cohort.

Other limitations of the presented study are the retrospective character and the moderate cohort size (*n* = 90). Also, the study was designed to match *n* = 50 patients with PVR to *n* = 40 patients without PVR, so taken together, patients with PVR are overrepresented. Furthermore, comparing patients’ outcomes from patients treated in the time frame 2012–2019 might be challenging since chemotherapy regimes have been improved over time, adjuvant chemotherapy as well as palliative therapy regimes. At least, overall survival will be improved for the patients treated more recently.

## 5. Conclusions

Overall, these data present PV/VMS resection and infiltration as a poor prognostic marker. This might fortify the hypothesis that tumors infiltrating the PV or even showing adherence to the PV are more aggressive tumor types. For future analyses, it should be recognized that parameters like venous infiltration or type of venous resection might play a relevant role when analyzing the effects of PVR.

## Figures and Tables

**Figure 1 biomedicines-11-03025-f001:**
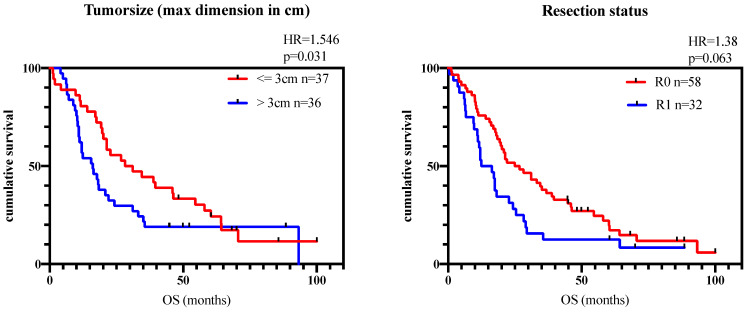
Overall survival (OS) depending on resection status and tumor size. R0—complete resection, R1 microscopically residual tumor in at least one organ margin/surface). Left: Patients with tumor size > 3 cm (blue line) show a significantly poorer OS compared to patients with tumor size < 3 cm (red line). Right: Patients with R0-resection status (red line) show a trend of better OS compared to patients with an R1-resection status (blue line, *p* = 0.059, not significant).

**Figure 2 biomedicines-11-03025-f002:**
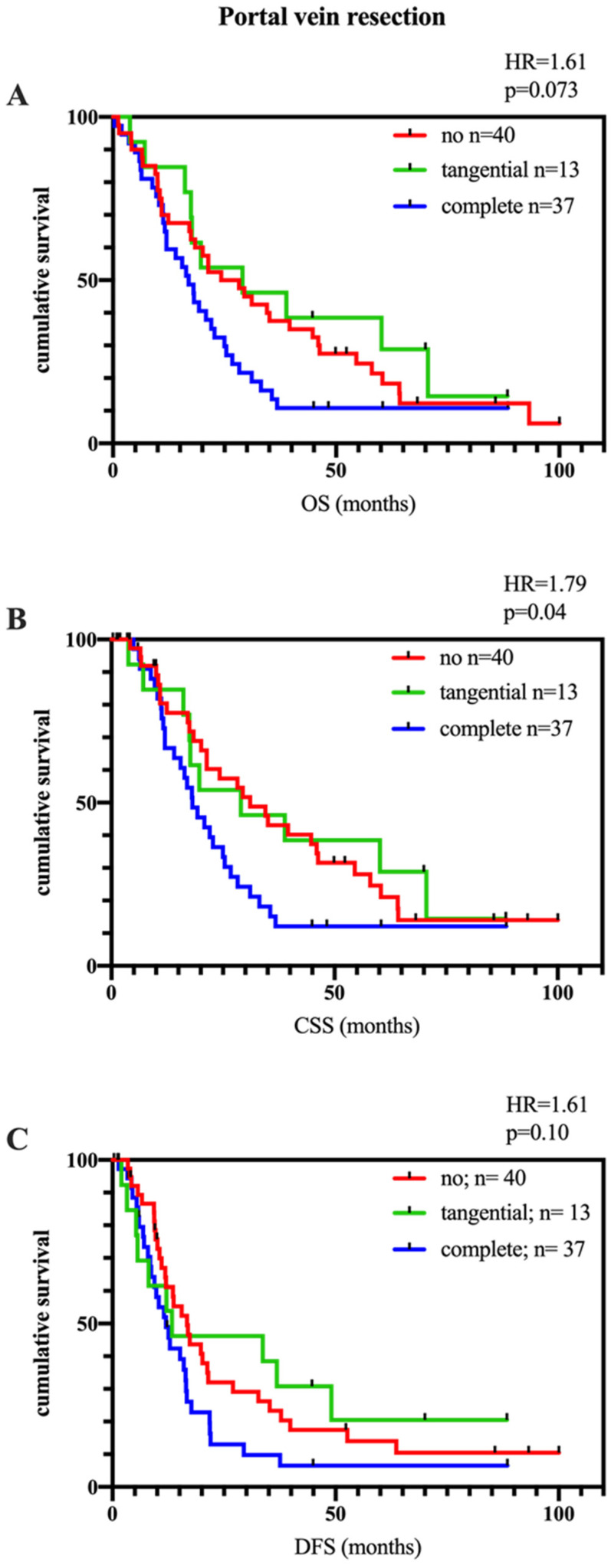
Long-term outcome of all included patients with PDAC after pancreatic head surgery, divided by performed complete PVR (blue line), tangential PVR (green line) and not-resected portal vein (red line). (**A**): Overall survival, OS, no statistically significant difference (*p* = 0.073, no vs. complete), (**B**): Cancer-specific survival, CSS, is significantly poorer for patients with complete PVR compared to non-PVR patients (*p* = 0.04, no vs. complete), (**C**): disease-free survival, DFS, no statistically significant difference between the groups.

**Figure 3 biomedicines-11-03025-f003:**
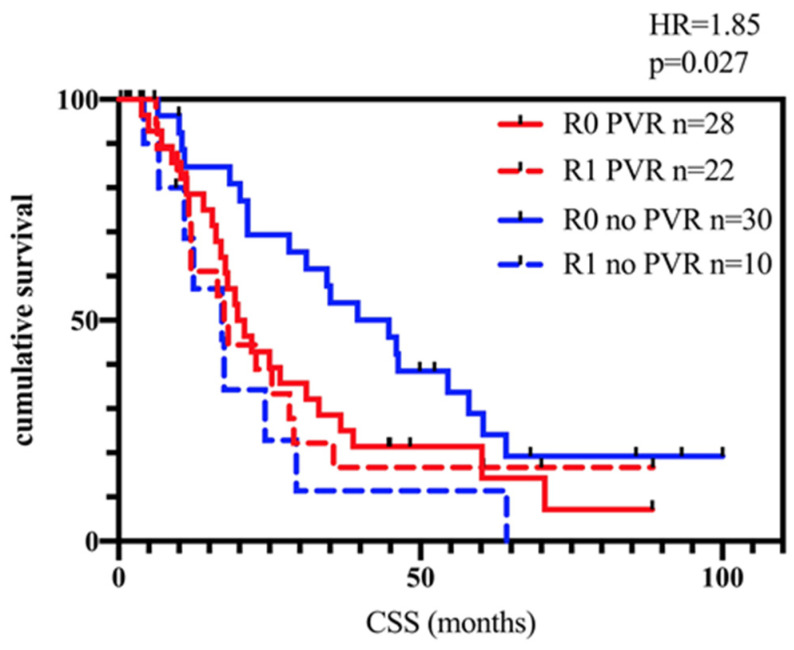
CSS of all patients, divided by R-state and whether they had a PVR during surgery. There is no significant difference between patients with R1-state without PVR (dashed blue line) and with PVR (dashed red line). Also, no significant difference was seen between R0-patients with PVR (red line) compared to R1-patients. Only R0 patients without PVR (blue line) showed significantly better CSS (*p* = 0.027).

**Figure 4 biomedicines-11-03025-f004:**
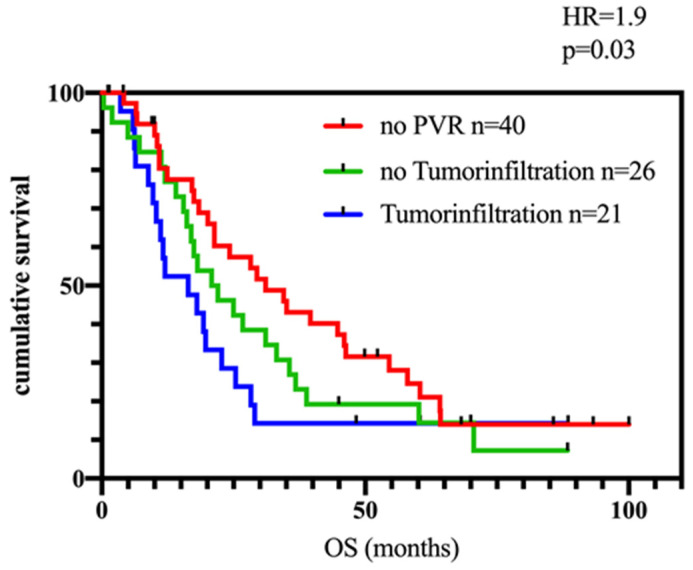
OS data for patients stratified by PVR depending on histopathological confirmed cancer infiltration in the PV. OS was significantly poorer in the group of patients with tumor infiltration of the portal vein (blue line) compared to those without any PVR (red line; *p* < 0.05). PVR without Tumorinfiltration (green line).

**Figure 5 biomedicines-11-03025-f005:**
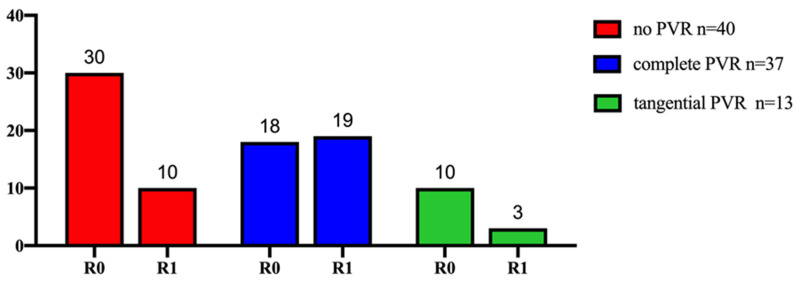
Comparison between complete (blue), tangential (green) and patients with no (red)-PVR patients concerning complete resection (R0 vs. R1). Histopathological complete resection was significantly lower in patients with complete PVR (Odds-Ratio = 3.17, *p* = 0.03).

**Table 1 biomedicines-11-03025-t001:** Clinical and histopathological data of enrolled patients. PPPD—pylorus preserving pancreatoduodenectomy, TP—total pancreatectomy, POPF—postoperative pancreatic fistula; DGE, delayed gastric emptying; PDAC, pancreatic ductal adenocarcinoma; CRM, circumferential resection margin.

		*n = 90*	*Percentage*	*PVR*	*non-PVR*
*Type of surgery*					
	PPPD	76	84.4%	41 (53.9%)	35 (46.1%)
	Whipple	6	6.7%	3 (50%)	3 (50%)
	TP	8	8.9%	6 (75%)	2 (25%)
*Portal vein resection*		50	55.6%		
	Wedge Resection	13	26%		
	Complete resection	37	74%		
*Patients with surgical complications*		23	25.6%		
	POPF	15	16.7%	6 (40%)	9 (60%)
	DGE	7	7.8%	2 (28.6%)	5 (71.4%)
	Postoperative bleeding	7	7.8%	3 (42.9%)	4 (57.1%)
	Leakage of bile duct anastomosis	4	4.4%	1 (25%)	3 (75%)
*Neoadjuvant treated patients*		7	7.8%	2 (28.6%)	5 (71.4%)
*Adjuvant treated patients*		63	70%	34 (68%)	29 (72.5%)
	FOLFIRINOX	5		5	0
	Gemcitabine	43		19	24
	Gemcitabine/Capecitabine	11		8	3
	Gemcitabine/nab-paclitaxel	3		1	2
	Capecitabine	1		1	0
*(y)pT-Stage*					
	T0	2	2.2%	0 (0%)	2 (100%)
	T1	4	4.5%	1 (25%)	3 (75%)
	T2	21	23.3%	15 (71.4%)	6 (28.6%)
	T3	63	70%	34 (54%)	29 (46%)
	T4	0	0%	-	-
*(y)pN-Stage*					
	N0	25	27.8%	11 (44%)	14 (56%)
	N1	56	62.2%	32 (57.1%)	24 (42.9%)
	N2	9	10%	7 (77.8%)	2 (22.2%)
*Grading*					
	G1	1	1.1%	0 (0%)	1 (100%)
	G2	65	72.2%	39 (60%)	26 (40%)
	G3	22	24.5%	11 (50%)	11 (50%)
	GX	2	2.2%	0 (0%)	2 (100%)
*Resection margin*					
	R0	58	64.4%	28 (48.3%)	30 (51.7%)
	CRM+	19	33.8%	13 (68.4%)	6 (31.6%)
	CRM−	24	41.4%	7 (29.2%)	17 (70.8%)
	CRMx	15	25.8%	8 (53.3%)	7 (46.7%)
	R1	32	35.6%	22 (68.8%)	10 (31.1%)

## Data Availability

The datasets used and/or analyzed during the current study are available from the corresponding author upon reasonable request.

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
