# Peer review of "Impact of Portal Vein Resection (PVR) in Patients Who Underwent Curative Intended Pancreatic Head Resection"

_biomedicines, 2023, doi:10.3390/biomedicines11113025_

Round 1

Reviewer 1 Report (Previous Reviewer 2)

Comments and Suggestions for Authors

Q1. This study is a retrospective analysis with a limited sample size (n=90 ), so its conclusions may not be generalizable.

Q2. The subgroup analysis showed that median OS in patients with RO resection was 22 months and R1 resection was 16.8 months in the previous version. Meanwhile, RO resection was 20.3 months and R1 resection was 14.2 months in this updated vision (P=0.063, but P=0.059 in Figure 1), with the median OS of all patients being 22 months. Still incorrect. In common sense, a median OS of 22 months cannot be achieved with R0 20.3 months and R1 14.2 months.

Q3. Figure 2A-C HR and P values were compared among 3 groups (no-PVR, tangential PVR vs complete PVR) or between 2 groups (no-PVR vs PVR). There is a lack of clarity in the presentation.

Author Response

Q1. This study is a retrospective analysis with a limited sample size (n=90 ), so its conclusions may not be generalizable.

We agree that the results of our study need to be validated in a larger patient cohort. 

Q2. The subgroup analysis showed that median OS in patients with RO resection was 22 months and R1 resection was 16.8 months in the previous version. Meanwhile, RO resection was 20.3 months and R1 resection was 14.2 months in this updated vision (P=0.063, but P=0.059 in Figure 1), with the median OS of all patients being 22 months. Still incorrect. In common sense, a median OS of 22 months cannot be achieved with R0 20.3 months and R1 14.2 months.

Response 2:

Thank you for the critical review on this point.  In order to make sure that all data is correct now in this revised version, we checked all numbers and survival rates listed In the following: 
median OS all R0 (n=58): 22 months
median OS all R1 (n=32): 14.4 months
median OS PVR R0 (n=28): 19.7 months
median OS PVR R1 (n=22): 14.2 months
median OS noPVR R0 (n=30): 34.5 months
median OS noPVR R1 (n=10): 14.7 months

(The rates in the last version were the one's of the PVR subgroup (R0 n=28, R1 n=22) and not the one's of all R0 (n=58) and R1 (n=32))

Q3. Figure 2A-C HR and P values were compared among 3 groups (no-PVR, tangential PVR vs complete PVR) or between 2 groups (no-PVR vs PVR). There is a lack of clarity in the presentation.

Response 3: The comparison is between the group of patients with no PVR and the group of patients with complete PVR. We added the information also in the figure legend to make the comparison more clear.

Reviewer 2 Report (Previous Reviewer 3)

Comments and Suggestions for Authors

The authors adequately responded to reviewers' comments. I do not have any additional comments for further revision.

Author Response

Thank you!

Reviewer 3 Report (New Reviewer)

Comments and Suggestions for Authors

thank you for allowing me to review this retrospective monocentric analysis measuring the impact of vein resection in pancreatic carcinological surgery. 

only patients with pancreatic adenocarcinoma were included to eliminate histological bias. a total of 50 patients with partial or total portal vein resection were included. why did the authors include only 40 patients without portal resection? given the inclusion period and the prospective database, it would be interesting to carry out a propensity score study including a larger number of control patients. this strategy would improve statistical power. 

In addition, the manuscript lacks tables describing the characteristics of each patient group. 

Author Response

only patients with pancreatic adenocarcinoma were included to eliminate histological bias. a total of 50 patients with partial or total portal vein resection were included. why did the authors include only 40 patients without portal resection? given the inclusion period and the prospective database, it would be interesting to carry out a propensity score study including a larger number of control patients. this strategy would improve statistical power. 

Thank you for this comment. We totally agree that a higher number of control patients would be desirable in general. However, we wanted to make sure that this is an independent collective of patients, whereas other patients undergoing pancreatic head resection at our department had already been included in previous studies (Dusch N, Oldani M, Steffen T, Kitz J, Koenig U, Azizian A, König A, Ströbel P, Beissbarth T, Ghadimi M, Gaedcke J. Intensified Histopathological Work-Up after Pancreatic Head Resection Reveals Relevant Prognostic Markers. Digestion. 2021;102(2):265-273. doi: 10.1159/000504648. Epub 2020 Jan 21. PMID: 31962319. 

We agree that our presented results should be followed by a prospective study including a propensity score since the current patient cohort is not large enough. 

In addition, the manuscript lacks tables describing the characteristics of each patient group. 

Meanwhile, we added further information about specific characteristics of each group into our revised manuscript as desired. If you could still not see these tables, could you please specify which characteristics would be needed?

Round 2

Reviewer 3 Report (New Reviewer)

Comments and Suggestions for Authors

the authors responded point by point to questions and comments designed to improve the quality of the manuscript

This manuscript is a resubmission of an earlier submission. The following is a list of the peer review reports and author responses from that submission.

Round 1

Reviewer 1 Report

Comments and Suggestions for Authors

This research article presents a German single-center retrospective study including 90 patients between 2012 and 2019 who underwent pancreatic head resection for pancreatic cancer with portal vein resection (n=50) or without (n=90). The objective of this study was to determine the relationship between venous resection and the oncologic prognosis of patients. This study provides evidence of nonconsensus and conflicting results between venous resection during pancreaticoduodenectomy and oncologic outcomes. 

Major criticisms

Regarding the study objective, the primary endpoint and study hypothesis are not clearly defined. 

The two groups to be compared (with or without venous resection) are not clearly presented and it is impossible to judge their homogeneity. 

The arbitrary choice of prognostic factors (tumor size, R0/R1 resection, adenopathy, venous resection or not...) and their associations do not seem to be adapted to the interpretation of the results. In order to find prognostic factors for survival, would it not have been more relevant to perform a uni and multivariate analysis including all these factors?

Similarly, in order to compare the populations with and without venous resection, it might also have been relevant to carry out a propensity score in order to erase as much as possible the differences between each group.

The inclusion period is long and there is no precision on the realization of an adjuvant chemotherapy which is validated at present for any pancreatic adenocarcinoma knowing that the protocols evolved during the last decade with a strong impact on the survival.

How many patients have had adjuvant chemotherapy and what protocol?

The interest of the results concerning tumor infiltration of the portal vein is not obvious and we do not understand what the author is trying to highlight except that tumor infiltration of the portal vein is a negative prognostic factor which is not found in other studies especially if the resection is R0.

Minor criticisms

1- Pancreatic surgery is a source of important post-op complications that can be increased by venous resection. These complications can have a negative influence on the ontological follow-up mainly if it is not possible to do adjuvant chemotherapy. Is it possible to have the complications in each group and their type?

2- It would have been interesting to determine if venous resection was planned preoperatively using for example the Ishikawa classification which has been shown to be effective in predicting the risk of positive margin and the need for venous resection. 

Similarly, there is no precision on the pre-therapeutic classification of the pancreatic tumor: resectable, border line or locally advanced.

What type of workup was performed preoperatively (CT scan, MRI...)? Is this morphological assessment standardized?

3- It is surprising to see that the resection margin is undetermined in 25.8% of cases, whereas we know that this result is essential for the oncological prognosis.

Furthermore, 35.6% of R1 resections were noted without it being possible to have the details according to whether or not the portal or mesenteric vein was resected.

Can the location of the microscopic residue be specified: venous bed, retro portal blade, posterior face of the head of the pancreas?

What is the protocol of the histopathological examination?

4- Among the 90 patients in the study, only 7 had neodajuvant chemotherapy. It is surprising that with 50 patients who had a venous resection that there are not more border line tumors for which neoadjuvant chemotherapy is recommended.

Can you clarify this?

Comments on the Quality of English Language

No comments

Reviewer 2 Report

Comments and Suggestions for Authors

In this study, Authors retrospectively analyzed the clinical data of 90 cases of pancreatic head cancer at the University Medical Centre Göttingen, 50 of which had portal vein resection (PVR), and analyzed the effect of PVR on the survival time of patients after surgery, the results suggested that patients with PVR were found to be significantly associated with poorer cancer-specific survival, regardless of the PVR resection types.

The review comments are as follows:

Q1. Results 3.2 showed median survival of all patients was 12 months, but subgroup analysis showed that patients with R0 resection was 22 months and R1 resection was 16.8 months, which may be incorrect.

Q2. Figure 1 Analysis of tumor size and overall survival time of patients after surgery, the number of patients in the two groups was 37 and 36 respectively, why were there 17 patients less?

Q3. It is suggested to add the clinical and histopathological data of PVR and no-PVR for group comparison, and analyze the factors affecting the efficacy of PVR. Meanwhile, if there were any differences of distant metastasis (liver and lung metastasis, etc.) local recurrence after surgery between the PVR group and no-PVR group.

Q4. It was mentioned that … Twenty one (44.7%) of 47 patients showed a positive infiltration whereas the majority (n=26, 55.3%) was resected without histopathological confirmed infiltration….., surgical indications for portal vein resection? were there any differences in OS, DFS, CSS between confirmed positive or negative infiltration?

Reviewer 3 Report

Comments and Suggestions for Authors

The authors presented the outcomes of pancreaticoduodenectomy with ant without PVR and showed worse prognosis in the patients with PVR probably due to higher R1 resection rate. However, the background was not fully compared between the patients with and without PVR, so it was difficult to draw clear conclusion which subgroup of patients could be good candidate for aggressive PVR. How the authors chose the type of PVR (wedge or sleeve) was also unclear.

Lots of similar studies have been published so far and I do not think this paper presented any novel finding.